# Impact of VR-Based Cognitive Training on Working Memory and Inhibitory Control in IDD Young Adults

**DOI:** 10.3390/healthcare12171705

**Published:** 2024-08-26

**Authors:** Maria João Trigueiro, Joana Lopes, Vítor Simões-Silva, Bruno Bastos Vieira de Melo, Raquel Simões de Almeida, António Marques

**Affiliations:** 1Laboratório de Reabilitação Psicossocial (LabRP), Centro de Investigação em Reabilitação (CIR), Escola Superior de Saúde (E2S), Polytechnic of Porto, 4200-072 Porto, Portugal; mjtrigueiro@ess.ipp.pt (M.J.T.); 10150299@ess.ipp.pt (J.L.); vds@ess.ipp.pt (V.S.-S.); bbm@ess.ipp.pt (B.B.V.d.M.); ajmarques@ess.ipp.pt (A.M.); 2Occupational Therapy Technical and Scientific Area, Santa Maria Health School, 4049-024 Porto, Portugal

**Keywords:** intellectual developmental disability, virtual reality, cognitive training, serious games, executive functions

## Abstract

Background: Young people with intellectual developmental disabilities have a persistent delay in the development of executive functions. Virtual reality (VR) is increasingly being used as a cognitive intervention tool, with significant effectiveness demonstrated in different types of populations. Methods: This pilot study aims to investigate the impact of a cognitive training program utilizing VR on young adults diagnosed with intellectual developmental disabilities (IDDs). The participants (N = 15) served as their own control group and were assessed three times: weeks 0, 8, and 16, with a rest period (0–8 weeks) and an intervention period (8–16 weeks). The assessments included measures of cognitive function provided by E-Prime^®^ (Version 3). Results: Overall, an improvement in working memory and inhibitory control was found after the intervention, but not in sustained attention. Conclusions: These findings suggest that VR-based cognitive training holds promise as an effective intervention for enhancing cognitive abilities in young adults with intellectual developmental disabilities. This study provides a foundation for future investigations into VR’s role in cognitive rehabilitation and its potential to support daily living skills and overall quality of life for individuals with IDDs. Further research is needed to explore the long-term effects and broader applicability of VR interventions.

## 1. Introduction

Intellectual developmental disabilities (IDDs) are neurodevelopmental disorders characterized by the presence of intellectual, functional, and adaptative deficits in conceptual, social, and practical domains [1]. IDDs manifest during the developmental period and generally persist throughout life, with different levels of cognitive impairment severity [1,2], being often associated with other developmental disorders such as cerebral palsy, autism spectrum disorder, Down syndrome, or fragile X syndrome [3].

Difficulties in functioning can be explained by problems in concentration, processing information, memory, or self-regulation, thus, compromising their autonomy and independence in daily life [1,4,5,6,7]. Prior research suggests that individuals with an intellectual developmental disability have a persistent delay in development and a slower rate of acquisition of executive functions [8,9]. These are higher-order cognitive mechanisms, which include working memory, processing speed, attentional control, planning, inhibitory control, solving problems that require decision-making processes for the selection of a functional response, and cognitive flexibility as a response to environmental contingencies [10,11,12,13,14,15,16,17]. It has been reported related to deficits in working memory, inhibitory control and verbal fluency [18], cognitive planning [11], processing speed [9], and attention and cognitive flexibility [9,11,18,19] in individuals with intellectual developmental disabilities. These deficits contribute to difficulties in solving intellectual challenges crucial for daily autonomy [20]. Cognitive training seems to be crucial for addressing these issues, as it aims to improve executive functions [21,22,23]. Traditional cognitive training involves several activities [24,25] but lacks real-time feedback, which limits its effectiveness [23]. For individuals with intellectual disabilities, technology primarily encompasses cognitive support tools, mainstream technologies, and supplemental communication aids. When these technologies are integrated into individual planning, they can significantly enhance daily life participation for adults with these conditions [26]. Newer technologies, such as computerized cognitive training, offer innovative interventions that are not only adaptable to individual performance but also capable of providing immediate feedback. However, the screen-based nature of such training may reduce its ecological validity and limit the transferability of skills to real-life situations [27]. Immersive and semi-immersive VR devices create a greater sense of presence by simulating real-life scenarios with real-time feedback, where users’ interactions with a virtual avatar induce cognitive and physical responses in their real bodies (embodied simulation), enhancing the feeling of ownership and immersion [28,29].

Virtual reality (VR) has gained popularity in neuroscience and as an intervention approach, proving to be effective for various deficits, especially in cognitive areas [15,21,27,29,30,31,32,33,34]. Immersive VR systems, utilizing head-mounted displays, provide interactive, embodied experiences with advantages such as non-invasiveness and real-time, controlled multisensory scenarios [27,35,36]. Immersive VR fosters a safe environment, promoting patient acceptance and calm skill practice [35,37,38]. It offers insights into brain activity, efficient performance feedback [27], and motivation through interactivity [39]. Also, immersive VR allows for the intervention to be more easily programmed, objective, and progressively graded [38,40], particularly in executive functions, serving as both an intervention and assessment tool in ecologically relevant conditions [17,29,32,39]. A 2021 systematic review [41] highlighted that VR is often chosen for its ability to individualize treatment, provide interactive experiences, and offer high ecological validity. VR’s capacity to simulate various representation modes (visual and auditory) and create realistic, adaptable environments was also noted. While all studies utilized VR for immersive learning, the specific advantages leveraged included simulating rare situations, ensuring safety, and making abstract concepts more tangible. The choice of technology typically depends on the skills being targeted.

Combining VR with serious games showed positive results in learning and skill improvement, as serious games enable goal-oriented operations within an entertaining environment [42,43]. Another systematic review [44] stated that digital interventions show promise for improving executive functions or basic cognitive skills, and commonly used tasks include games and videos, with positive reinforcement and frequent repetition enhancing effectiveness. While some short-term studies report benefits, longer interventions generally provide more consistent results, demonstrating that digital methods may be more effective than traditional approaches.

As effective cognitive interventions for IDDs should prioritize motivation, task complexity, grading, and acquisition assessment, it seems promising that greater sensory immersion might enhance cognitive processing, suggesting that virtual environments may stimulate executive functions in IDDs [12]. Consistently, previous literature suggests the potential of serious games with VR as a rehabilitation tool for individuals with intellectual developmental disabilities [32,37]. Nonetheless, existing VR cognitive training studies predominantly focus on patients with traumatic brain injury, stroke, mild cognitive impairment, and dementia, with limited attention to individuals with ID [23,40,45]. Furthermore, while VR interventions for physical and daily life skills are explored [38], research on executive function development in individuals with an intellectual developmental disabilities is scarce [5,46]. Therefore, this study aims to investigate the effects of cognitive training using immersive VR on executive functions, specifically working memory, sustained attention, and inhibitory control, in young adults with intellectual developmental disabilities.

## 2. Materials and Methods

This study employed a quasi-experimental design with a one-group pre-test–post-test structure (Figure 1). The study follows the TREND Statement Checklist for the reporting quality of nonrandomized evaluations of behavioral and public health interventions [47]. Participants served as their own control group, undergoing assessments before (two times, with an eight-week period without any intervention in between), and after the intervention [48,49].

### 2.1. Participants

A convenience sampling method was used, and 15 individuals attending services at the Centro de Atividades e Capacitação para a Inclusão—APACI were selected for the study. This institution, located in Barcelos, functions as a center for activities and training for community inclusion.

Inclusion criteria were (a) young adults diagnosed with mild or moderate intellectual developmental disabilities, (b) aged between 18 and 35 years old (c) ability to understand instructions given in Portuguese, (d) previous experience using mouse and gamepads for gaming, and (e) expressed motivation to participate in this study. Participants with (a) health conditions that could interfere with the quality of the participant’s participation (e.g., epilepsy, severe vision and hearing impairments, and motor deficits), (b) behavioral issues that could impede engagement, (c) difficulty understanding the game mechanism, and (d) concurrent similar intervention were excluded.

### 2.2. Instruments

A sociodemographic questionnaire was used, covering age, sex, literacy, level of the IDD, and previous VR experience. Executive function variables were assessed using tests provided by E-Prime^®^, a software package designed for psychological experiments and cognitive science [50,51,52]. E-Prime^®^ was operated on a computer running Windows 10, Intel^®^ Core™ i7-6500U processor, 15-inch screen, and a USB mouse. Tests for visual-spatial working memory, sustained attention, and inhibitory control were conducted at an average distance of 50 cm from the participant’s field of vision.

The process of participant randomization is described in the flowchart prepared according to the CONSORT guidelines [53], presented in Figure 2.

#### 2.2.1. Corsi Block-Tapping Task

The Corsi Block-Tapping Task (CBTT) is a test that measures visuospatial short-term and working memory. In this test, nine squares appear on a blue screen and light up in yellow, one by one, in a variable sequence. After the stimulus presentation, participants must reproduce the sequence by clicking on each of the squares that turned yellow. The test starts with a simple sequence task that increases or decreases in complexity (varying between two and eight elements) based on participants’ performance [54,55]. The test used 20 sequences. A correct answer was considered when all the numbers in the sequence were right; therefore, the number of correct answers was used as a performance measure. The score varies between 0 and 20, and a higher score means a better performance.

#### 2.2.2. Simple Reaction Time Task

The Simple Reaction Time (SRT) is a test for sustained attention and processing speed [56,57]. In this test, a single star-shaped stimulus is repeatedly presented at the same location on the screen, and participants must press the “1” key as quickly as possible. The time interval between stimuli varies throughout the task [52]. The test used 60 trials. A correct answer was considered when it was provided after the stimulus presentation; therefore, the number of correct answers was used as a performance measure. The score varies between 0 and 60, and a higher score means a better performance.

#### 2.2.3. Stop Signal Task

The Stop Signal Task (SST) is a test designed to assess inhibitory control, involving a go signal requiring a response and a stop signal requiring a cancellation of a response [58,59]. Participants are instructed to quickly respond to a left or right arrow presentation using, respectively, the “q” and “p” keyboard keys (go task). Periodically, stimuli appear surrounded by a red light during which participants must withhold their action of pressing any key (stop task). Feedback is provided after each attempt (Psychology Software Tools, https://cambridgecognition.com/stop-signal-task-sst/ (accessed on 25 August 2024)). Given the participants’ characteristics, keyboard keys were labeled to match the direction of the response arrows—the “q” key with a left arrow (<) and the “p” key with a right arrow (>). The test used 151 trials. A correct answer was considered when the response was coherent with the stimuli direction; therefore, the number of correct answers was used as a performance measure. The score varies between 0 and 151, and higher score means a better performance.

These tasks were chosen based on their established appropriateness for measuring the specific cognitive outcomes under investigation [60,61]. Furthermore, these tasks were well-suited to the characteristics of our study population, given their ease of administration and comprehension by participants.

### 2.3. Procedures

This study received approval from the Ethics Committee of Escola Superior de Saúde do Politécnico do Porto (CE0109C/2022), and all procedures conformed to the principles in the Declaration of Helsinki [62]. After approval by the APACI institution, participants were selected during March 2023, and the first moment of assessment occurred. A second assessment occurred eight weeks later, without any intervention, to establish a baseline (Figure 2). Before the intervention moment, all participants underwent the benchmark session of the Enhance VR Games, receiving explanations regarding the objectives and controls. Participants also had a training session to explore the VR equipment—a benchmark session. The intervention started in May 2023, took place at the APACI institution, and included 24 sessions. The third and final assessment was conducted in July 2023. Data were collected in paper format for the sociodemographic questionnaire, and digital format, through E-Prime tasks, for the executive functions’ assessment. All data were coded to maintain confidentiality and will be stored for 10 years by the principal investigator [63].

To promote adherence to the intervention, when the intervention was finished, the researchers provided detailed information about the study and explained the benefits of their participation. A close follow-up was given: the schedule was provided in a timely manner according to the participant’s availability, and ensuring that the session was rescheduled in case of absences, the session would be rescheduled at a time convenient for both parties [64].

#### 2.3.1. Intervention Program: Enhance VR—Virtuleap

For cognitive training, three games available on the Enhance VR platform were used. Enhance VR is an app accessible either through a subscription or for research purposes consisting of a library of cognitive exercises developed by Virtuleap [27], which is a health and education VR startup. We aim to elevate the cognitive assessment and training industry with the help of emerging technologies such as virtual reality and artificial intelligence. Games were accessed through a Meta Quest 2 head-mounted display, Qualcomm snapdragon 835 processor, 4 GB RAM, 128 GB internal memory, 1400 × 1600 resolution per eye in pixels, with a refresh rate of 72 Hz, and motion controllers. The intervention protocol, informed by findings in the literature, consisted of twenty-four 20-minute sessions, three sessions per week for eight consecutive weeks. Games were played in the same sequence—React, Memory Wall, and Whack-A-Mole—and mainly in a standing position. Sitting position was allowed if participants felt tired, but only in the Memory Wall game, as it requires less movement. The same researcher was present in all sessions. A brief overview of each game is provided next.

#### 2.3.2. React

The React game (Figure 3a) was designed to train task switching and response inhibition skills and is based on the mechanisms of the Wisconsin Card Sorting Test [65] and the Stroop Task mechanisms [66]. The player needs to categorize approaching objects according to their shape and color, throwing them into two portals, which only accept matching objects. During the game, players need to adapt to dynamic contexts, as the portals can change their position and required objects during the levels. The difficulty increases by introducing distractor objects that must be ignored [27].

#### 2.3.3. Memory Wall

The Memory Wall game (Figure 3b) trains short-term visuospatial memory and was inspired by the Visual Patterns Test [67]. Players need to memorize the positions of illuminated cubes that appear for three seconds, in a three-dimensional grid in their field of vision, and then reproduce the pattern. Task difficulty increases with each level, depending on the size of the grid and the number of cubes [27].

#### 2.3.4. Whack-A-Mole

The Whack-A-Mole game (Figure 3c) focuses on sustained attention and was inspired by the Psychomotor Vigilance Test [68]. Players need to hit moles that appear at random intervals and holes before they disappear. Players need to react as quickly and accurately as possible. The difficulty increases as speed increases, and multiple moles can rise simultaneously [27].

### 2.4. Data Analysis

Data were exported to IBM SPSS Statistics (Version 28.0) for statistical analysis [69], considering a 0.05 significance level for all performed tests [70]. Descriptive statistics were used to characterize the sample, namely mean (x) and standard deviation (sd), for continuous or discrete variables, and frequencies (N; %) for nominal or ordinal data. The normality of variables was assessed through the Shapiro-Wilk test or the examination of data distribution using threshold criteria for skewness and kurtosis, aiming for values less than |2.0| and |9.0|, respectively [71]. One-way repeated-measures ANOVAs were employed to compare pre- and post-test conditions. Sphericity was tested using Mauchly’s test, with the Huynh–Feldt correction applied when this assumption was not met and the epsilon was higher than 0.57. In cases where this criterion was not met, the Greenhouse-Geisser correction was utilized [71]. The Bonferroni test was used as a post-hoc measure to determine where the actual differences between the three evaluation moments are located.

## 3. Results

This sample consisted of 15 participants (Table 1), aged between 22 and 34 years old (mean age = 28.07 ± 3.97), and most were males (66.70%). Participants had mild (53.30%) or moderate (46.70%) IDD levels, and eight (53.30%) were illiterate. None had previous experience with VR.

Results for the CBTT, SRT, and SST in the three moments of assessment (Table 2) show that there were statistically significant differences in the scores of the working memory (p_CBTT_ = 0.001) and inhibitory control (p_SST_ = 0.043), suggesting that the group’s performance improved with the intervention. The attention test was not significantly different over time (p_SRT_ = 0.101).

A post-hoc test for score differences between the moments of assessment (Table 3) shows that in CBTT (working memory) and SST (inhibitory control), there were no differences when comparing the first and second moments (p_CBTT_ = 1.000; p_SST_ = 1.000), but when both are compared with the moment after the intervention the differences are statistically significant in working memory (test 1 vs. test 3—p_CBTT_ = 0.004; test 2 vs. test 3—p_CBTT_ = 0.002) and inhibitory control (test 1 vs. test 3—p_SST_ = 0.010; test 2 vs. test 3—p_SST_ = 0.039).

The analysis of the influence of IDD levels on the results of the assessment results (Table 4) shows that there was a statistically significant difference between IDD levels in working memory (p_CBTT_ = 0.002) and inhibitory control (p_SST_ = 0.032). The interaction between IDD level and sustained attention does not have significant values.

A post-hoc test for score differences between the three moments of assessment when the level of ID is taken into account (Table 5) shows that in CBTT (working memory) and SST (inhibitory control), there are no differences between the first and second moments (p_CBTT_ = 0.450; p_SST_ = 0.786), but the differences are statistically significant when both moments were compared with the third assessment in both levels of IDD (test 1 vs. test 3—p_CBTT_ = 0.002; p_SST_ = 0.032; test 2 vs. test 3—p_CBTT_ = 0.001; p_SST_ = 0.009).

## 4. Discussion

This study aimed to assess the effectiveness of an immersive VR cognitive training intervention, using serious games, on working memory, sustained attention, and inhibitory control in young adults with intellectual developmental disabilities. Overall, an improvement in working memory and inhibitory control was found, but not in sustained attention, both in the whole group and considering IDD level. Although not in all the variables, the positive result in executive functions is in line with previous studies that have used similar cognitive training interventions [37,42,46,72,73]. In fact, despite the literature being scarce, a recent systematic review of the effects of computerized task-based cognitive training programs in a game environment proved to be helpful for people with intellectual developmental disabilities [46]. They reported multiple studies with significant positive effects across different cognitive domains, such as visual working memory and attention, especially in adults with intellectual developmental disabilities [46]. Furthermore, Giachero, Quadrini, Pisano, Calati, Rugiero, Ferrero, Pia, and Marangolo [37] divided 14 subjects into three groups according to different levels of IDD and found a greater performance in executive functions tasks—attention and short and long-term spatial memory—in all groups after the treatment, especially in the mild IDD group. Thus, using computerized cognitive training appears to be an effective strategy for improving the executive functions of young people with intellectual developmental disabilities. Specifically, immersive VR training in rehabilitation programs seems to further provide the advantage of practicing sensory-motor, cognitive, behavioral, and adaptive functions in a safe, close-to-real-world simulation. Positive changes in working memory following the intervention were found. As there were no differences between the first and second moment (i.e., before the intervention), it is reasonable to conclude that these changes were caused by the Enhance VR games. These results are consistent with previous studies that, equally, reported working memory improvements after a computerized cognitive training program for people with intellectual developmental disabilities. Roording-Ragetlie, Spaltman, de Groot, Klip, Buitelaar, and Slaats-Willemse [73] examined the impact of CogMed Working Memory Training on children with intellectual developmental disabilities in a blind randomized trial, observing improvements in working memory tasks in the group undergoing cognitive training. Another study [72] found that verbal short-term memory improved in teenagers with mild to borderline intellectual developmental disabilities, after a 5-week intervention, three 6-minute computerized cognitive training sessions per week. Kim and Lee [74] employed a 24-session game-based cognitive training program (30-minute sessions, biweekly, for three months) with children with intellectual developmental disabilities and discovered that the experimental group improved in working memory performance. 

Significant improvements in inhibitory control following the intervention were found, although to a lesser extent than working memory. As far as the authors know, there is little research on inhibitory control intervention for people with intellectual developmental disabilities. McGlinchey et al. [75] conducted a quasi-experimental study to investigate the influence of a cognitive training program on executive functions in people with Down syndrome who had mild to moderate intellectual developmental disabilities. The intervention included 20 min of Scientific Brain Training Pro, 5 days a week, for 8 weeks. Post-intervention findings showed significant gains in inhibition control and working memory.

Inhibitory control was reported in the literature to have a medium to large deficit in people with intellectual developmental disabilities, particularly in behavioral inhibition and interference control [76], which are believed to be more deliberate types of inhibition. According to the inhibition taxonomy proposed by Nigg [77], these two subtypes of executive inhibition are defined as the “processes for intentional control or suppression of responses in the service of higher-order goals” (p. 238). In Danielsson’s study [78], inhibitory control responses were much lower in the IDD group compared to the other two groups—with identical chronological age and identical mental age. These difficulties may have to do with the fact that they had to recruit other cognitive skills linked to mental age, such as working memory (for example, keeping the rules of the task constantly updated) to carry out the task. This seems to be consistent with our own findings—where working memory and inhibitory control improved together—and earlier research conducted by Thorell et al. [79], which suggested that these two components of executive functions are interrelated, with the functioning of one influencing the functioning of the other. Thus, working memory training may lead to gains in inhibitory tasks and vice versa, enhancing the possibility of improvement in these components. Thus, this relationship can potentially explain our findings, where these two variables improved together following the intervention program, but not sustained attention.

No significant changes were found in sustained attention between pre- and post-treatment assessment. As with inhibitory control, research on sustained attention in people with intellectual developmental disabilities is scarce, but our findings are consistent with a previous randomized control study that aimed to assess the efficacy of a computerized attention training program in children with intellectual developmental disabilities [80]. They concluded that, despite observed improvements in selective attention, none were observed regarding sustained attention.

Several studies found that people with intellectual developmental disabilities have a lower performance in reaction time [81,82,83] compared to controls with typical development but not in visual sustained attention [82,84,85]. This means that the absence of improvement in our sample could have been influenced by the motor component of the task that was used to assess sustained attention. Indeed, it has been shown that individuals with intellectual developmental disabilities present longer premotor time [86], which could influence the motor component of reaction time. Vogt et al. [87] also reported that the SRT remained unchanged following a self-selected 30-minute running exercise in individuals with intellectual developmental disabilities. However, several other authors have reported improvements in reaction time after programs that include physical exercise, such as the games chosen for this intervention. For example, Ringenbach et al. [88] reported that the reaction time improved in individuals with Down syndrome after assisted cycling at 80 revolutions per minute but remained unchanged after voluntary cycling at the participant’s self-selected rate. The authors explained this result based on the difference of pace, as in the assisted cycling intervention, individuals with intellectual developmental disabilities cycled at a rate 49.3% greater than the mean self-selected rate in the voluntary cycling intervention. Chen and Ringenbach [89] showed that 20 min of walking on a treadmill at a moderate intensity improved reaction time in individuals with Down syndrome. Affes, Borji, Zarrouk, Sahli, and Rebai [81] suggested that low to moderate running exercises improve reaction time in people with intellectual developmental disabilities and that low-intensity exercise, rather than moderate, could be more appropriate to enhance reaction time. Therefore, this discrepancy with our results might be due to the difference in exercise intensity, which could be insufficient to produce any reaction time improvement. The design of studies with longer or more intensive interventions could change these results.

An improvement in working memory and inhibitory control independent of IDD level was found, but performance differences between IDD levels have been reported, where children [90,91,92], adolescents, and adults [37,76] with mild intellectual developmental disabilities had fewer problems in executive functions domains than those with moderate IDDs. Nonetheless, the fact that no statistically significant differences in performance were found is consistent with other studies. Giachero, Quadrini, Pisano, Calati, Rugiero, Ferrero, Pia, and Marangolo [37] reported that all participants showed a better performance in a VR gardening task (twice a week for 14 weeks), regardless of IDD level. Actually, their sample also performed better in working memory and inhibitory control after the program sessions. However, Giachero, Quadrini, Pisano, Calati, Rugiero, Ferrero, Pia, and Marangolo [37] found that the three IDD groups improved equally in attention and short- and long-term spatial memory tasks, concluding that the VR videos trained not only the participants’ gardening skills but also had a significant impact on tasks requiring executive functions, attentional, and spatial skills, that were closely related to the observed procedures. Perhaps this variability in results stems from the inherent heterogeneity of IDDs.

Using VR-based interventions targeting executive functions such as working memory, sustained attention, and inhibitory control in individuals with intellectual developmental disabilities is not new but is not extensively explored in the literature. Only in the past decade has it re-emerged as a promising adjuvant treatment strategy for cognitive rehabilitation [31,93], so there is still interest in continuing studies that explore different approaches, populations, and results. This study used an innovative platform—Enhance VR—which uses various cognitive training games accessed through a head-mounted display. It allows for a higher level of immersion and a strong sense of presence, given the simultaneous motor, visual, and proprioceptive systems integration, which is effective for enhancing motor and cognitive skills [21,94,95] in several populations. Also, other studies suggest that VR-based approaches are stimulating and allow more immediate feedback on performance, promoting more motivation and adherence to treatment [23,96].

This study has limitations worth mentioning. First, the convenience sample was small, which prevented it from being divided into experimental and control groups. However, as we used the group as its own control, it was possible to compare the first and second moments (without intervention) with the third moment (after the intervention). That given, most likely, the changes seen were due to the intervention program, as it was the only change introduced during this period. An argument in favor of the program efficacy is related to the fact that the skills of people with intellectual developmental disabilities tend to progress slower in time when compared to typically developing people (for a longitudinal study, see [97]. Hence, the improvement might be due to our 24-session program. On the other hand, an argument against this is that people with intellectual developmental disabilities tend to have fewer skill-based activities when compared to typically developing people (for an observational study, see [98]). Hence, the improvement we saw might be due simply to an added training activity. Either argument is in favor of the efficacy of this program—that we argue that could be related to its VR-based design, as discussed above, and consistent with a recent meta-analysis that reported the effectiveness of serious games on social and cognitive skills of children with intellectual developmental disabilities [99]. 

Additionally, the use of immersive VR in cognitive training could present challenges that must be considered when interpreting the results of this study. One significant challenge is the potential for motion sickness, which can occur due to sensory conflicts experienced in the VR environment [100]. This can lead to discomfort, nausea, and dizziness, potentially limiting the effectiveness and usability of VR for some participants—which did not occur in this study. Furthermore, age-related effects may influence how individuals interact with and adapt to VR technology: younger participants may be more adept at navigating and engaging with VR environments, while older individuals or those unfamiliar with digital interfaces might face greater difficulties [101]. These factors were carefully considered when selecting an immersive VR application for our study, as they may impact both the engagement levels and outcomes of the cognitive training. The study protocol was designed to mitigate these challenges as much as possible, but their presence remains an important consideration for future research and application of VR-based interventions.

Replicating our study, or other VR-based intervention, with larger samples and a control group is recommended. Also, a follow-up assessment after the end of the intervention was not carried out, and study designs that address a follow-up assessment are recommended. Thus, it is not possible to know whether the effects obtained immediately after the intervention were maintained in the sample subjects and, even more so, whether they were successfully applied in their daily performance, demonstrating whether there was generalization of the results acquired. To gain insights into the real-world applicability of the skills acquired through the intervention, instruments such as the Vineland Adaptive Behavior Scale could be considered [102]. This tool measures adaptive behavior and functioning in daily life, offering a broader perspective on skill application. However, such instruments often require longer intervention periods to detect meaningful changes due to their less sensitivity to short-term outcomes. Therefore, this study’s initial focus was only on evaluating immediate changes in cognitive capacity.

VR technology has become increasingly accessible and cost-effective, making it a viable option for cognitive interventions [103]. The initial costs of VR equipment have decreased significantly in recent years, and the availability of user-friendly platforms has expanded, reducing the barriers to implementation. Compared to traditional therapeutic methods, VR offers a unique, immersive experience that can be tailored to individual needs, potentially enhancing engagement and outcomes. While the upfront investment in VR technology may still be higher than some conventional methods, the ability to deliver personalized and scalable interventions presents a cost-benefit advantage, allowing for saving resources associated with in-person therapy sessions. Moreover, the potential for remote and home-based VR applications can further offer a more flexible and economical solution for ongoing cognitive training [104]. This increasing accessibility and the potential for broader application support the rationale for integrating VR into cognitive interventions, particularly for populations that may benefit from more innovative and engaging therapeutic approaches.

## 5. Conclusions

The use of VR as a therapeutic approach for individuals with intellectual developmental disabilities is still uncommon and requires further investigation. This study presents promising results, indicating that VR interventions can potentially enhance cognitive performance in this population. However, it is important to acknowledge that the goal of any therapeutic intervention is to facilitate the transfer of newly acquired skills to real-world applications, including daily living activities and community participation. Our study demonstrated the efficacy of VR-based training in improving specific cognitive outcomes within a controlled experimental setting, but it did not assess whether these cognitive gains translate into meaningful benefits in everyday life. To address this gap, future research should focus on evaluating the real-world applicability of VR interventions. Longitudinal studies tracking participants’ progress in their daily routines and social interactions post-intervention will be crucial. Additionally, investigating the impact of VR-based cognitive training on practical aspects such as job performance, social skills, and overall quality of life will provide a more comprehensive understanding of the intervention’s effectiveness.

Given the complex nature of IDDs, characterized by multiple limitations and compromised functionality, VR offers an innovative tool for immersive and highly customizable training. It holds the potential to create unprecedented, simulation-based interventions within a safe and controlled environment. To fully validate the preliminary findings of this study and explore the long-term effects and practical implementation of VR-based interventions, larger-scale studies with clinical populations are warranted.

## Figures and Tables

**Figure 1 healthcare-12-01705-f001:**
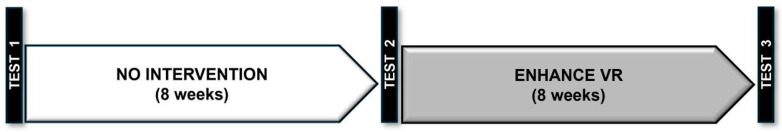
One-group pre-test–post-test structure.

**Figure 2 healthcare-12-01705-f002:**
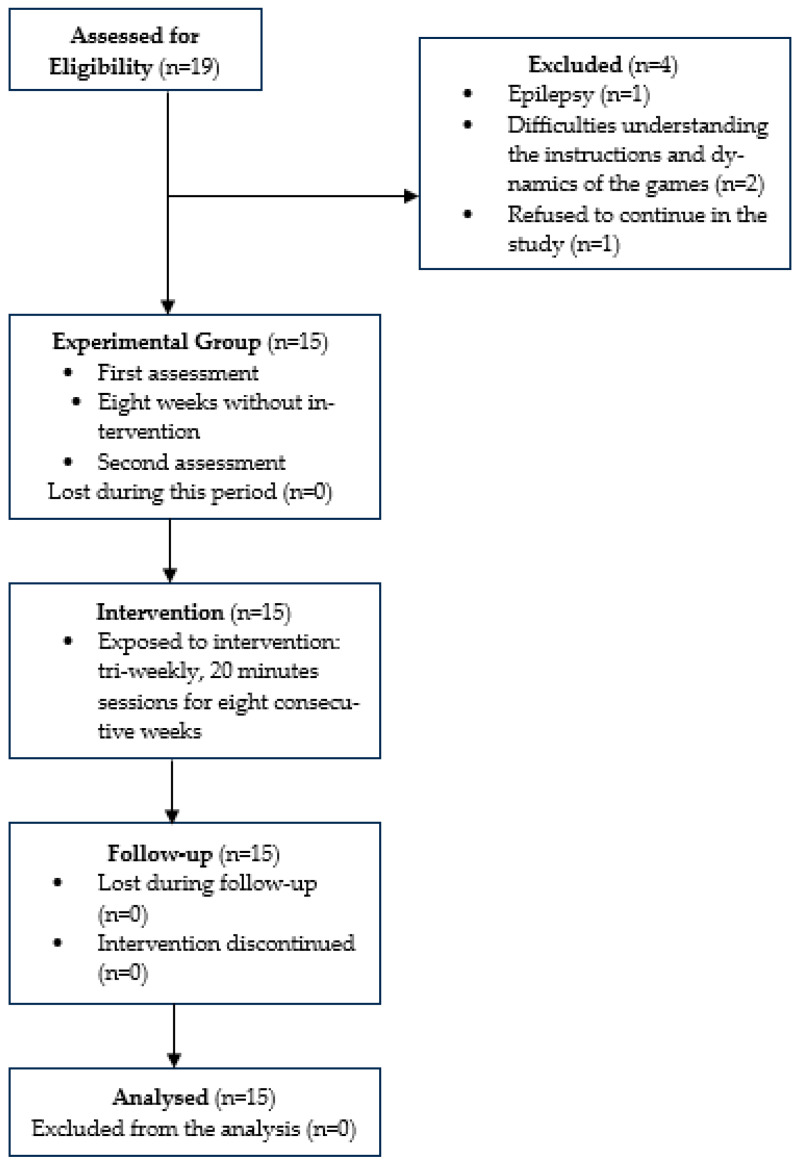
CONSORT diagram of study design.

**Figure 3 healthcare-12-01705-f003:**
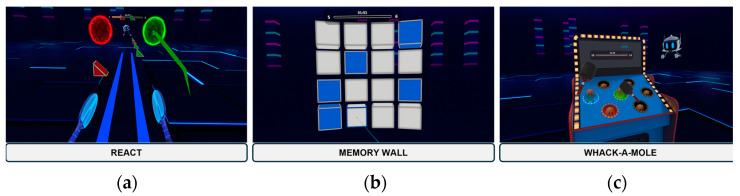
React (**a**), Memory Wall (**b**), and Whack-A-Mole (**c**).

**Table 1 healthcare-12-01705-t001:** Sample’s sociodemographic characteristics.

		x ± SD	N (%)
Age (years)		28.07 ± 3.97	
Gender	Male		10 (66.70)
	Female		5 (33.30)
Level of IDD	Mild		8 (53.30)
	Moderate		7 (46.70)
Literacy	Yes		7 (46.70)
	No		8 (53.30)
Previous experience with VR	Yes		0 (0.00)
	No		15 (100.00)

IDD—intellectual developmental disability; VR—virtual reality; x—mean; SD—standard deviation; N—absolute frequency; %—relative frequency.

**Table 2 healthcare-12-01705-t002:** Score differences in the three moments of assessment.

	Test 1 x ± SD	Test 2 x ± SD	Test 3 x ± SD	*p*-Value	Power
CBTT	11.53 ± 2.03	11.07 ± 2.54	13.93 ± 1.91	0.001 *	0.960
SRT	55.86 ± 5.41	53.47 ± 11.21	57.73 ± 3.57	0.101	0.373
SST	90.40 ± 22.76	90.87 ± 23.89	99.53 ± 27.58	0.043 *	0.545

CBTT—Corsi Block-Tapping Task; SST—Stop Signal Task; SRT—Simple Reaction Time; x—mean; SD—standard deviation; *p*-value—Within-subjects *p*-value; * *p* < 0.05).

**Table 3 healthcare-12-01705-t003:** Score differences between moments of assessment for CBTT and SST.

	CBTT		SST	
	Mean Difference	*p*-Value	Mean Difference	*p*-Value
Test 1 vs. Test 2	0.467	1.000	−0.467	1.000
Test 1 vs. Test 3	−2.400	0.004 *	−9.133	0.010 *
Test 2 vs. Test 3	−2.867	0.002 *	−8.667	0.039 *

CCBTT—Corsi Block-Tapping Task; SST—Stop Signal Task; * *p*-value—pairwise comparisons Bonferroni; * *p*-value < 0.05).

**Table 4 healthcare-12-01705-t004:** Score differences between moments of assessment according to IDD level.

	IDD Level	Test 1	Test 2	Test 3	*p*-Value ^a^	*p*-Value ^b^	Power ^a^	Power ^b^
CBTT	Mild	11.63 ± 2.26	11.25 ± 3.28	14.50 ± 2.33	0.002 *	0.418	0.949	0.121
	Moderate	11.43 ± 1.90	10.86 ± 1.57	13.29 ± 1.11				
SRT	Mild	55.25 ± 5.44	56.88 ± 4.58	56.88 ± 4.58	0.112	0.818	0.352	0.055
	Moderate	56.57 ± 5.71	57.00 ± 3.32	58.71 ± 1.80				
SST	Mild	99.63 ± 7.46	98.75 ± 8.16	103.38 ± 10.00	0.032 *	0.168	0.605	0.272
	Moderate	79.86 ± 7.98	81.86 ± 8.72	95.14 ± 10.69				

*p*-value ^a^—within-subjects *p*-value; *p*-value ^b^—interaction *p*-value; power ^a^—Within-subjects; power ^b^—Interaction; CCBTT—Corsi Block-Tapping Task; SST—Stop Signal Task; SRT—Simple Reaction Time; IDD—intellectual developmental disability; * *p*-value < 0.05).

**Table 5 healthcare-12-01705-t005:** Score differences between moments of assessment considering intellectual disability level.

	Level of IDD			
	CBTT		SST	
	Mean Difference ± sd	*p*-Value ^a^	Mean Difference ± sd	*p*-Value ^a^
Test 1 vs. Test 2	0.47 ± 0.61	0.450	0.56 ± 2.03	0.786
Test 1 vs. Test 3	−2.37 ± 0.61	0.002 *	9.52 ± 3.95	0.032 *
Test 2 vs. Test 3	−2.84 ± 0.69	0.001 *	8.96 ± 2.93	0.009 *

CCBTT—Corsi Block-Tapping Task; SST—Stop Signal Task; IDD—intellectual developmental disability; *p*-value ^a^—pairwise comparisons Bonferroni; * *p*-value < 0.05.

## Data Availability

Data supporting the findings of this study are available from the corresponding author upon reasonable request.

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
