# Peer review of "Impact of VR-Based Cognitive Training on Working Memory and Inhibitory Control in IDD Young Adults"

_healthcare, 2024, doi:10.3390/healthcare12171705_

Round 1

Reviewer 1 Report

Comments and Suggestions for Authors

Dear authors, the work shows its importance and originality, here are some comments and observations on your work:

For the affiliations of authors belonging to the same institution, only mention it once and share the label (a) of the affiliation.

In the correspondence author section, only the mail should be included.

The abstract should be worked on more, not only including the results and conclusions sections in isolation, but also achieving an integration of the work in perspective in this section.

In the introduction, where the importance of virtual reality in cognitive areas is addressed, it is not clear which serious games are those that can be implemented for this type of cognitive interventions, it is important to highlight the results of previous research using these serious games even if they are not in their immersive variant, as well as the cons of the use of virtual reality in this type of patients and in general, such as sickness motion, and the results by age groups, since this study focuses on young people of a certain age. This will help to substantiate why the virtual reality application was chosen to be immersive.

Specify whether Enhace VR - Virtuleap is free and whether the developers of the virtual platform propose the intervention protocol or what criteria were followed to propose 20-minute sessions.

It is also important to mention if during the sessions the patients presented any type of dizziness, sickness motion, headache or any reaction.

Did the evaluation that was applied include any comments from the patients that could be mentioned in the results? To know what the patients' perception was.

Please include images of the implementation of the immersive virtual reality application. 

In the conclusions it is important to comment on whether the results obtained are comparable with other types of interventions, justifying the implementation of your proposal, especially in terms of the cost-benefit ratio.

Expand this statement, since it is understood that this study allows visualizing this future work: However, it should be recalled that the goal of an intervention is to transfer the newly acquired skills to daily living activities and community participation, which was not covered in this experiment.

Considering a section on future work could help specify all that is envisioned that could potentially help the patients who were treated.

Regards

Comments on the Quality of English Language

Minor editing of English language required.

Author Response

Dear authors, the work shows its importance and originality, here are some comments and observations on your work:

For the affiliations of authors belonging to the same institution, only mention it once and share the label (a) of the affiliation. In the correspondence author section, only the mail should be included.

Response: Thank you for this guidance. We believe this will be adjusted during proof reading.

The abstract should be worked on more, not only including the results and conclusions sections in isolation, but also achieving an integration of the work in perspective in this section.

Response: We have revised the abstract accordingly.

In the introduction, where the importance of virtual reality in cognitive areas is addressed, it is not clear which serious games are those that can be implemented for this type of cognitive interventions, it is important to highlight the results of previous research using these serious games even if they are not in their immersive variant, as well as the cons of the use of virtual reality in this type of patients and in general, such as sickness motion, and the results by age groups, since this study focuses on young people of a certain age. This will help to substantiate why the virtual reality application was chosen to be immersive.

Response: Thank you for this observation. We have included references to prior research. Additionally, we have discussed the limitations and challenges of using virtual reality, including motion sickness and age-related effects in the discussion section.

Specify whether Enhace VR - Virtuleap is free and whether the developers of the virtual platform propose the intervention protocol or what criteria were followed to propose 20-minute sessions.

Response: We’ve added this information in the manuscript.

It is also important to mention if during the sessions the patients presented any type of dizziness, sickness motion, headache or any reaction.

Response: Participants did not present any of those symptoms. We’ve added this information in the manuscript.

Did the evaluation that was applied include any comments from the patients that could be mentioned in the results? To know what the patients' perception was.

Response: Although we did not ask for any qualitative feedback, participants' informal perceptions of the intervention were positive.

Please include images of the implementation of the immersive virtual reality application. 

Response: We are ethically prohibited from using photos of participants to protect their privacy and confidentiality. However, we have included images of the games used in the immersive virtual reality application to provide visual context for the intervention.

In the conclusions it is important to comment on whether the results obtained are comparable with other types of interventions, justifying the implementation of your proposal, especially in terms of the cost-benefit ratio.

Response: We have expanded the conclusions to compare our findings with other interventions and discuss the cost-benefit implications, thereby providing a rationale for the proposed VR intervention.

Expand this statement, since it is understood that this study allows visualizing this future work: “However, it should be recalled that the goal of an intervention is to transfer the newly acquired skills to daily living activities and community participation, which was not covered in this experiment.”

Response: We have adjusted the conclusion section to better address your suggestion.

Considering a section on future work could help specify all that is envisioned that could potentially help the patients who were treated.

Response: Thank you for your suggestion. Although we did not create a section for this point, we tried to develop future work in the manuscript.

We have made the necessary revisions based on your valuable feedback. Thank you for your thorough review and constructive suggestions. All the changes in the manuscript are highlighted in blue.

Reviewer 2 Report

Comments and Suggestions for Authors

The results of the design used indicate the need to consider some evidence from the literature in similar studies, for example it is necessary to have an entry assessment, this in order to establish at least a baseline, then it has been found that at least periods of continuous training of between 12 and 20 hours are required to evaluate the progression, it is also important to consider the possibilities of transferring improvements in skills to other cognitive skills and daily life. The authors indicate periods of time of development of the tests but do not indicate the intensity in accumulated hours of training, the minimum and maximum times recorded and the reactions of the participants (in some studies alterations of orientation, stability, balance have been found), this is especially relevant taking into consideration that in Table 1, it is indicated that the 15 participants had no previous experience with VR, the authors do not indicate if there was a phase of training and learning of performance in virtual environments. 

Author Response

The results of the design used indicate the need to consider some evidence from the literature in similar studies, for example it is necessary to have an entry assessment, this in order to establish at least a baseline, then it has been found that at least periods of continuous training of between 12 and 20 hours are required to evaluate the progression, it is also important to consider the possibilities of transferring improvements in skills to other cognitive skills and daily life. The authors indicate periods of time of development of the tests but do not indicate the intensity in accumulated hours of training, the minimum and maximum times recorded and the reactions of the participants (in some studies alterations of orientation, stability, balance have been found), this is especially relevant taking into consideration that in Table 1, it is indicated that the 15 participants had no previous experience with VR, the authors do not indicate if there was a phase of training and learning of performance in virtual environments. 

Response: Thank you for your thorough review. Regarding the study design, we have focused mainly on the participants characteristics (young people with IDD) and the recommendations that already exists in the literature - session duration and program length. Concerning the study design, we have chosen to make a quasi-experimental one group pretest-posttest design (and not a time series design as written, which was a mistake and has since been fixed) since this kind of study measures scores before and after the treatment and compare them.

Please see these references:

  • Gravetter, F. J., & Forzano, L.-A. B. (2019). Quasi-Experimental and Single-Case Experimental Designs. In Research Methods for the Behavioral Sciences (pp. 333-372). SAGE Publications, Inc.
  • Reichardt, C. S., Storage, D., & Abraham, D. (2023). Quasi-Experimental Research. In A. L. Nichols & J. Edlund (Eds.), The Cambridge Handbook of Research Methods and Statistics for the Social and Behavioral Sciences: Volume 1: Building a Program of Research (pp. 292–313). chapter, Cambridge: Cambridge University Press.
  • Price, P., Jhangiani, R., & Chiang, I. (2015). Research Methods of Psychology – 2nd Canadian Edition. Victoria, B.C.: BCcampus. Retrieved from https://opentextbc.ca/researchmetho.
  • Johnson, R. B., & Christensen, L. B. (2019). Educational Research: Quantitative, Qualitative, and Mixed Approaches. Boston, MA: Allyn and Bacon.
  • Handley, M. A., Lyles, C. R., McCulloch, C., & Cattamanchi, A. (2018). Selecting and Improving Quasi-Experimental Designs in Effectiveness and Implementation Research. Annual review of public health, 39, 5–25. https://doi.org/10.1146/annurev-publhealth-040617-014128.

Concerning the other data (accumulated hours of training, the minimum and maximum times recorded and the reactions of the participants), the intervention protocol consisted of twenty-four sessions, conducted three times per week over a span of eight consecutive weeks. Each session followed the same sequence of Enhance VR games: React, Memory Wall, and Whack-A-Mole. The first session of each game serves as a benchmark session, which is longer in duration and is used to establish each user's performance plateau. This benchmark session provides a baseline from which subsequent training sessions are derived. The training sessions that follow are shorter and begin at the last level achieved in the previous session. The duration of each session is fixed as follows: Whack-a-Mole: Benchmark session: 3 minutes; Training sessions: 1.5 minutes: React: Benchmark session: 6 minutes; Training sessions: 2 minutes; Memory Wall: Benchmark session: 6 minutes; Training sessions: 2 minutes. Before starting gameplay, users complete a self-paced tutorial during their first interaction with the games. This tutorial can be accessed again before each one of the game sessions either upon user request or as selected by the instructor.

Also, we agree that the transfer of skills is a crucial aspect of cognitive interventions. We have expanded the discussion to consider how improvements observed in the VR-based tasks might transfer to other cognitive skills and daily life activities. We know that there are some instruments to measure the functioning and adaptive behavior, such as Vineland Adaptive Behavior Scale, giving us a broader perspective how the skills were used on their daily live. However, we also know that this type of instruments requires longer periods of interventions due to the less sensitivity of the outcomes measured. Thus, we’ve decided to, first, test the changing in capacity, and then, the results were promising, we would adapt the intervention – making it longer – to assess the performance.

We’ve added information about the phase of training and learning, which in fact occurred. All the changes in the manuscript are highlighted in blue.

Reviewer 3 Report

Comments and Suggestions for Authors

Your study offers a novel and significant exploration of the VR-based cognitive training for young adults with intellectual developmental disabilities (IDD),  hence contributing valuable insights to this area of research which is still in its emergent phase. However, please consider the following advice to strengthen your manuscript.

For instance, the introduction would benefit from a more thorough discussion of earlier research that was specifically related to VR interventions for people with IDD. This could be achieved by including more references to recent systematic reviews or meta-analyses in this field.

Given the circumstances and limitations (such as the small sample size), the quasi-experimental design employed in this study is appropriate. The lack of a control group, however, is a limitation that needs to be brought up more in the discussion. Even though the authors use the group as its own monitoring system, adding a comparison with an outside control group—even if it's just acknowledged as a limitation—would strengthen the findings that they draw from the research.

In the methodology, please consider providing justification for the selection of the particular cognitive tasks (Corsi Block-Tapping Task, Simple Reaction Time Task, and Stop Signal Task) as opposed to other potential assessments methods. It would also be beneficial to have more information on how the VR intervention was customized for the participants with IDD.

Tables are used appropriately to summarize the data and present the results in a clear and concise manner. The results of the statistical analyses are easily interpreted, and the statistical analyses are effectively reported.

Although the results generally support the conclusions, they could be strengthened by clearly outlining the study's limitations, especially its small sample size and lack of long-term follow-up.

Comments on the Quality of English Language

English language fine, only minor edits needed in sentence structure and correcting minor grammatical errors.

Author Response

Your study offers a novel and significant exploration of the VR-based cognitive training for young adults with intellectual developmental disabilities (IDD), hence contributing valuable insights to this area of research which is still in its emergent phase. However, please consider the following advice to strengthen your manuscript.

Response: Thank you very much for your thoughtful review.

For instance, the introduction would benefit from a more thorough discussion of earlier research that was specifically related to VR interventions for people with IDD. This could be achieved by including more references to recent systematic reviews or meta-analyses in this field.

Response: We appreciate this suggestion and we have incorporated additional references related to VR interventions for individuals with IDD.

Given the circumstances and limitations (such as the small sample size), the quasi-experimental design employed in this study is appropriate. The lack of a control group, however, is a limitation that needs to be brought up more in the discussion. Even though the authors use the group as its own monitoring system, adding a comparison with an outside control group—even if it's just acknowledged as a limitation—would strengthen the findings that they draw from the research.

Response: We acknowledge the limitation of not having an external control group in our study and we have stated that on the discussion section.

In the methodology, please consider providing justification for the selection of the particular cognitive tasks (Corsi Block-Tapping Task, Simple Reaction Time Task, and Stop Signal Task) as opposed to other potential assessments methods. It would also be beneficial to have more information on how the VR intervention was customized for the participants with IDD.

Response: Thank you for this observation. We have added a detailed rationale for selecting the Corsi Block-Tapping Task, Simple Reaction Time Task, and Stop Signal Task. We have used these tasks due to several reasons. First, there are the cognitive tasks that we are available in our research lab, and second, they were scientific appropriate for the measurement of these outcomes. Moreover, these three tests align to the characteristics of our population since they were easier to apply and comprehend.

Please see these references:

  • Handley, M. A., Lyles, C. R., McCulloch, C., & Cattamanchi, A. (2018). Selecting and Improving Quasi-Experimental Designs in Effectiveness and Implementation Research. Annual review of public health, 39, 5–25. https://doi.org/10.1146/annurev-publhealth-040617-014128.
  • Guo, Z., Gong, Y., Lu, H., Qiu, R., Wang, X., Zhu, X., & You, X. (2022). Multitarget high-definition transcranial direct current stimulation improves response inhibition more than single-target high-definition transcranial direct current stimulation in healthy participants. Frontiers in neuroscience, 16, 905247.
  • Xu, Y., Zhang, W., Zhang, K., Feng, M., Duan, T., Chen, Y., Wei, X., Luo, Y., & Ni, G. (2022). Basketball training frequency is associated with executive functions in boys aged 6 to 8 years. Frontiers in human neuroscience, 16, 917385.
  • Duschek, Stefan, de Guevara, Cristina Muñoz Ladrón, Serrano, María José Fernández, Montoro, Casandra I., López, Santiago Pelegrina, Reyes del Paso, Gustavo A., Variability of Reaction Time as a Marker of Executive Function Impairments in Fibromyalgia, Behavioural Neurology, 2022, 1821684, 9 pages, 2022. https://doi.org/10.1155/2022/1821684.
  • Duschek, Stefan, de Guevara, Cristina Muñoz Ladrón, Serrano, María José Fernández, Montoro, Casandra I., López, Santiago Pelegrina, Reyes del Paso, Gustavo A., Variability of Reaction Time as a Marker of Executive Function Impairments in Fibromyalgia, Behavioural Neurology, 2022, 1821684, 9 pages, 2022. https://doi.org/10.1155/2022/1821684.
  • Edgin, J. O., Pennington, B. F., & Mervis, C. B. (2010). Neuropsychological components of intellectual disability: the contributions of immediate, working, and associative memory. Journal of intellectual disability research : JIDR, 54(5), 406–417. https://doi.org/10.1111/j.1365-2788.2010.01278.x.
  • Arnett, A.B., Flaherty, B.P. A framework for characterizing heterogeneity in neurodevelopmental data using latent profile analysis in a sample of children with ADHD. J Neurodevelop Disord 14, 45 (2022). https://doi.org/10.1186/s11689-022-09454-w.

No customization was made.

Tables are used appropriately to summarize the data and present the results in a clear and concise manner. The results of the statistical analyses are easily interpreted, and the statistical analyses are effectively reported.

Response: We appreciate the positive feedback on the use of tables and the clarity of our statistical analyses.

Although the results generally support the conclusions, they could be strengthened by clearly outlining the study's limitations, especially its small sample size and lack of long-term follow-up.

Response: We have revised the results and discussion sections to more clearly outline the study’s limitations, including the small sample size and the absence of long-term follow-up. We have discussed how these limitations might impact the interpretation of the findings and suggest ways to address them in future research.

We have incorporated all your suggestions to enhance the clarity, rigor, and overall quality of the manuscript. Thank you for the valuable feedback. All the changes in the manuscript are highlighted in blue.

Round 2

Reviewer 1 Report

Comments and Suggestions for Authors

Corrections have been made and the authors' comments have been addressed.

Best regards